# Comparison of the Immunogenicity and Safety of Three Enhanced Inactivated Poliovirus Vaccines from Different Manufacturers in Healthy Korean Infants: A Prospective Multicenter Study

**DOI:** 10.3390/vaccines8020200

**Published:** 2020-04-26

**Authors:** Byung Ok Kwak, Sang Hyuk Ma, Su Eun Park, Seon Hee Shin, Kyung Min Choi, Taek-Jin Lee, Byung Wook Eun, Young Jin Hong, Dong Hyun Kim

**Affiliations:** 1Department of Pediatrics, Hallym University Kangnam Sacred Heart Hospital, Seoul 07441, Korea; qquack00@hanmail.net; 2Department of Pediatrics, Changwon Fatima Hospital, Changwon 51394, Korea; msh6517@hanmail.net; 3Department of Pediatrics, Pusan National University School of Medicine, Busan 50612, Korea; psepse@naver.com; 4Department of Pediatrics, Hallym University Dongtan Sacred Heart Hospital, Hwaseong 18450, Korea; shshin@hallym.or.kr; 5Department of Pediatrics, CHA University Gangnam Medical Center, CHA University School of Medicine, Seoul 06135, Korea; ckm2001@hanmail.net; 6Department of Pediatrics, CHA University Bundang Medical Center, CHA University School of Medicine, Seongnam 13496, Korea; bjloveu@hanmail.net; 7Department of Pediatrics, Eulji University School of Medicine, Eulji University Hospital, Seoul 01830, Korea; acet0125@hanmail.net; 8Department of Pediatrics, Inha University School of Medicine, Incheon 22332, Korea; hongyjin@inha.ac.kr

**Keywords:** immunogenicity, safety, inactivated poliovirus vaccine, infants

## Abstract

The enhanced inactivated poliovirus vaccine was first introduced in 2002, and several inactivated poliovirus vaccines are licensed in Korea. Reliable data by a prospective study on the immunogenicity and safety of the inactivated poliovirus vaccines in Korean infants are required. Normal healthy infants aged 6–12 weeks received three doses of the vaccine (IPVAX™, Imovax Polio™ or Poliorix™) in intervals of 2 months. Neutralizing antibody (NTAb) titers were measured before and 4–6 weeks after three-dose primary vaccination. Immunogenicity was evaluated by seroconversion rates and geometric mean titers obtained by analyzing NTAb titers. Local and systemic adverse events were recorded during 7 days after each vaccination. A total of 150 infants were included: 40 in IPVAX™, 52 in Imovax Polio™, and 58 in Poliorix™. The seroconversion rates for the group vaccinated with IPVAX™ were 100% in types 1, 2 and 3, while those of Imovax Polio™ were 98.1%, 96.2%, 96.2% and those of Poliorix™ were 98.3%, 100%, 100%, respectively. In all groups, injection site redness and irritability were the most common local and systemic adverse events. Neither serious adverse events nor adverse events above grade 2 were reported throughout the study. The currently used inactivated poliovirus vaccines was demonstrated to be safe and immunogenic in healthy Korean infants.

## 1. Introduction

Since the poliovirus vaccine was introduced in the 1950s, the incidence of paralytic polio has been markedly reduced [1]. While it has been largely eradicated in most developed nations, occasional cases are reported among the unimmunized, which has been attributed to immigration. In developing nations, however, polio patients are still being reported [2]. In Korea, the old inactivated poliovirus vaccine (oIPV) for injection was introduced in 1962, and the oral poliovirus vaccine (OPV) was introduced in 1965. Both were used for vaccinations until 1975, and then only the OPV was used. Polio cases have decreased to 0.1 per 100,000 persons since the introduction of the poliovirus vaccine, and no new cases have been reported since the five reported cases in 1983 [3].

The enhanced-potency inactivated poliovirus vaccine (eIPV), which was developed for injection by van Wesel in 1978, uses the same virus strains as the oIPV [4]. However, the vaccine’s capacity for antibody formation was improved through greater antigenicity. In Korea, three different vaccination schedules were used in 2002: IPV injection alone, oral live vaccine alone, or injection and oral vaccine combination. Since 2004, only eIPV is given three times (at 2 months, 4 months, and 6 months) to complete the primary series, which is followed by one booster at 4–6 years [2]. Some European countries, including Sweden and Finland, have been using the eIPV since the late 1980s because of its superior antigenicity and safety [5]. However, studies of immunogenicity and safety of eIPV in Korean infants and children are scarce. Kim et al. conducted one study, but it was limited to a single product evaluation [6]. Currently, eIPVs from five different manufacturers are distributed in Korea, and combination vaccines including eIPV were introduced and used also. Therefore, a study of the immunogenicity and safety of eIPV from different manufactures currently available for Korean infants and children is imperative

## 2. Materials and Methods

### 2.1. Study Design and Participants 

This study was a multicenter and open-label clinical trial conducted in Korea between May 2009 and April 2011. The study protocol was approved by the institutional review board (IRB) at each participating hospital. The approved IRB number from Inha University Hospital IRB for this study was IUH-IRB 09-1605. Initially, written informed consent was obtained from the parents or legal guardians of the participants. Upon completion of physical screening and interviews in accordance with the clinical trial design, a subject number [ID] was given to those who met the inclusion criteria. All participants were healthy 6–12-week-old infants who were born after a 35- to 42-week gestational period with birth weights of 2500 g or above.

### 2.2. Study Vaccines and Administration

The 5 different poliovirus vaccines approved for use in Korea are Imovax polio™, Poliorix™, IPVAX™, Kovax-polio PF™, and NexPoly™. IPVAX™, Kovax-polio PF™, and NexPoly™ were all imported from Netherlands Vaccine Institute (NVI). As these are imported in bulk and then distributed under different names, only IPVAX™ was selected for this trial. Thus, immunogenicity and safety of the Imovax polio™, Poliorix™, and IPVAX™ vaccines were evaluated through IPVAX™.

eIPV was injected intramuscularly or subcutaneously at 2, 4, and 6 months in the left thigh according to the manufacturer’s instructions. Concomitantly, DTaP vaccine was administered in the right thigh. After each injection, the participants were closely monitored for 30 min for any serious adverse events, such as anaphylaxis.

### 2.3. Immunogenicity Assessment

Blood samples (5 mL) were collected from each participant pre-vaccination and 4 to 6 weeks after completion of the primary vaccination to measure neutralizing antibody titers, and immunogenicity was evaluated with the seroconversion rate and the geometric mean titer (GMT) [5,7]. 

The blood sample was centrifuged to separate the serum, and was stored in a −20 or −70 °C freezer until the antibody titer test. All serum samples were sent to the Virology Department, Research Institute for Tropical Medicine, Philippines, and neutralizing antibodies against poliovirus types 1, 2 and 3 were measured with the following procedure. Samples were tested with poliovirus types 1, 2, and 3 at dilutions ranging from 1:4 to 1:4096. A positive control, back titration, and 3 samples were added to each plate. Serum samples were diluted 1:4 (200 µL of serum, 600 μL of mem-s). After 30 min of inactivation at 56 °C, 10 tubes were prepared and labeled with an appropriate dilution ratio ranging from 1:8 to 1:4096. With a pipette, 400 μL of mem-s was transferred to the tubes. From the 1:4 inactivated sample, 400 μL was transferred to the 1:8 tube with a pipette, and double dilution up to 1:4096 was performed. Fifty microliters of the 100 tissue culture infective dose 50% (TCID50) poliovirus antigens 1, 2, and 3 was prepared. To wells A12-H12, 100 μL of 2% Minimal Essential Medium (MEM) was added, and 50 μL of the diluted samples was added to the wells. Fifty microliters of the diluted control was added to the well, and 50 μL of the poliovirus 1, 2, and 3 antigen suspension 100 cell culture infectious dose 50% (CCID50) was added to each plate except for the back titration wells. Next, 100 μL of CCID50, 50 CCID50, 25 CCID50, or 1 CCID50 was added to the marked wells, which were incubated at 36 °C for 3 h. One-hundred microliters of the RD-A cell suspension (1.8 × 105) was added. Plates were sealed with a plate sealer and incubated at 36 °C for 5 days, during which the plates were read every day. The antibody titer was expressed as the highest dilution ratio of the serum at which the cytopathic effects of polioviruses 1, 2, and 3 were not present. Seroconversion was defined as antibody titers of 1:8 or higher [8,9]. 

### 2.4. Safety Assessment

Solicited local (pain, redness, swelling, and induration) and systemic (fever, crying, irritability, poor appetite, drowsiness, vomiting, and diarrhea) adverse events were recorded by the parents or legal guardians for 7 days in a diary card, and the causal relationship between the adverse events (AEs) and the vaccine was assessed by the investigator at each visit. Unsolicited AEs were recorded up to 4 weeks after vaccination. Serious adverse events (SAEs) were recorded throughout the study and followed until symptoms disappeared. Adverse events were assigned grades: 0 = none, 1 = mild (asymptomatic or mild symptoms not affecting activities of daily living), 2 = moderate (AEs limiting activities of daily living), and 3 = severe (AEs preventing activities of daily living). 

### 2.5. Statistical Analyses

The sample size was calculated for the primary outcome to assess seroconversion rate of eIPV with the following parameter: alpha of 10%, delta of 0.05, assuming 96% seroconversion rates for all three types and 20% drop-out rate of participants. The required sample size was 53 in each group.

Demographic characteristics such as age and sex were described by means and standard deviations or frequencies and proportions. Seroconversion rates and GMTs were calculated with 95% confidence intervals (CIs). The number of AEs, frequencies, and proportions were calculated. The chi-square test or ANOVA were used to examine differences among the study groups. A *p* value < 0.05 was considered as statistically significant. All statistical analyses were performed using SPSS software version 24 (IBM Corp., Armonk, NY, USA).

## 3. Results

### 3.1. Study Population

Among 168 enrolled participants, 46 subjects received IPVAX™, 60 subjects received Imovax polio™, and 62 subjects received Poliorix™. The 150 (89.3%) participants who completed the study included 40 with IPVAX™, 52 with Imovax polio™, and 58 with Poliorix™. Baseline demographics were compared across the study groups (Table 1). 

### 3.2. Immunogenicity 

Of the 150 participants at 4 to 6 weeks after primary vaccination, overall seroconversion rates against poliovirus types 1, 2, and 3 were above 96% in all three groups, and no differences were found between the three groups. (Table 2). The seroconversion rates of the neutralizing antibody after three doses of primary vaccination in IPVAX™, Imovax polio™, and Poliorix groups were 100%, 98.08%, and 98.28%, respectively, against poliovirus type 1 (*p* = 0.6889); 100%, 96.15%, and 100% against poliovirus type 2 (*p* = 0.1481); and 100%, 96.15%, and 100% against poliovirus type 3 (*p* = 0.1481). 

At baseline, GMTs of neutralizing antibodies against poliovirus types 1, 2, and 3 were distributed between 4.96 and 7.89 in three study groups. After the primary vaccination, the GMTs of antibodies in IPVAX™, Imovax polio™, and Poliorix groups were 97.01, 78.17, and 102.00, respectively, against poliovirus type 1; 163.14, 144.31, and 181.02 against poliovirus type 2; and 226.76, 218.16, and 268.53 against poliovirus type 3. 

### 3.3. Safety

The incidence of AEs was similar between three groups (Table 3). There were no differences between three groups in the occurrence of solicited local AE. Although the high incidences of solicited systemic AEs to Polirix™ following the second and third doses were statistically significant, no AEs above grade 2 were observed.

The most common solicited local AE was injection site redness, reported for 45.00% of infants in the IPVAX™ group and 37.93% of infants in the Poliorix™ group. In the Imovax polio™ group, pain was the most common solicited local AE, reported for 46.15% of infants. No significant difference was observed between the vaccine groups for local AE. The most common solicited systemic AE was irritability, reported for 80.00%, 84.62%, and 81.03% of infants in the IPVAX™, Imovax polio™, and Poliorix™ groups, respectively. Irritability was the most common systemic symptom considered by the investigator to be related to vaccination. Unsolicited AEs were reported for 70.00%, 55.77%, and 63.79% of infants in the IPVAX™ (Daewoong Pharmaceutical Co., Seoul, Korea), Imovax polio™ (Sanofi Pasteur Ltd., Lyon, France), and Poliorix™ groups (GlaxoSmithKline Biologicals, Brentford, United Kingdom), respectively. Of them, upper respiratory infection was the most common reported unsolicited AE in all groups. Unsolicited AEs possibly related to vaccination were reported for 2.5%, 1.92%, and 1.72% of infants, respectively. No SAEs were reported in this study.

## 4. Discussion

The results of this study show that all eIPVs distributed in Korea induced protective levels of antibodies and seropositive levels after three doses of primary vaccination according to the Korean national immunization program (NIP). Safety profiles were clinically acceptable, and no safety issues were found.

Since the poliovirus vaccine was first introduced in the 1950s, cases of paralytic polio have declined rapidly [1]. In 1988, WHO reinforced routine vaccination against poliovirus in endemic regions with the goal of eradicating poliomyelitis worldwide by the year 2000 [10]. Consequently, the number of polio cases has markedly decreased; the 35,000 cases reported in 1988 decreased by 99% in 2000 to less than 3500 cases, and a wild type 2 poliovirus case has not been reported since 1999. In 2008, 1655 polio cases were reported, while less than 1000 cases were reported in 2010 [11]. However, these cases developed in regions where there had been no previous incidence, indicating the plan did not proceed as designed. Thus, the effort to eradicate wild type poliovirus continues. 

According to a study conducted on Korean polio patients from 1962 to 1964, 70% were less than 3 years old, with one-year-old infants accounting for the majority [6]. IPV was first distributed in Korea in 1962, and OPV was added in 1965. Since the introduction of the vaccines, the cases of polio have decreased to 0.1 per 100,000 persons, and the fatality rate has decreased to 0.1–4.3%. The five cases of polio that were reported in 1983 were the last known cases, and no new patients have been reported to date [6]. In 2000, the WHO Western Pacific Region, including Korea, was declared free of indigenous poliovirus. This certifies the absence of poliovirus for at least 3 years and the presence of a well-established surveillance system that effectively detects and reports all cases of acute paralytic polio associated with vaccination [12]. Korea introduced a vaccine-associated paralytic polio reporting system in 1988, and a single case of vaccine-associated paralytic polio has been confirmed to date [11]. 

IPV is a vaccine in which poliovirus types 1, 2, and 3 are generated through cell culture and inactivated by formalin. In early development, simian virus contamination was an issue because the vaccine viruses were produced by a primary culture method with monkey renal cells. Since the 1970s, human diploid cells or Vero cells are used for culturing viruses, thus eliminating the risk of simian virus contamination [8]. With advances in cell culture techniques, vaccine viruses can be produced in large quantity, and eIPV with enhanced immunogenicity was developed through greater antigenic content. Some European nations have used eIPV as their primary vaccination since the 1980s, and the US has been using eIPV only since 2000 [11,12]. 

Compared to IPV, OPV has several advantages, including superior immunogenicity, ease of vaccination through oral administration, and blocking the transmission of wild type viruses through intestinal immunity [13]. Thus, it has been used in most countries to prevent polio infection. However, with the decline in wild type poliovirus infection, vaccine-related paralysis following OPV administration has become a bigger issue. In addition, because vaccine-related polio can develop in immunocompromised patients, IPV, as opposed to OPV, is the preferred vaccination choice in countries in which polio has been eradicated. 

Five different eIPVs have been approved and distributed in Korea since 2004, the viruses of which were cultured in human diploid cells or Vero cells [11]. Each dose (0.5 mL) contains 40D of Mahoney strain (type 1), 8D of MEF-1 strain (type 2), and 32D of Saukett strain (type 3). They were inactivated by formaldehyde and contain a small amount of neomycin, streptomycin, and polymyxin B. The preservative used was 2-phenoxyethanol.

eIPV has greatly improved the immunogenicity against polioviruses compared to oIPV. While the immunogenicity is not higher following the initial dose, the protection antibodies against all three poliovirus types are raised by 90% following the second dose and by 99% following the third dose. In one US study, the antibody titer was above 1:100–1:1000 or 1:1000 when the three-dose vaccination completed 6 or 12 months after birth, respectively [14]. The antibody titer increased when the interval between the second and third vaccination was prolonged. According to 30 studies on two-dose vaccination, the antibody seroconversion rates for poliovirus types 1, 2, and 3 were 89–100%, 92–100%, and 70–100%, respectively. The antibody response of three-dose vaccination was better than after three-dose vaccination. Although the immune responses were acceptable when vaccinated at 3, 4, and 5 or 2, 3, and 4 months, superior immunogenicity was found when infants were vaccinated at 2, 3, and 4 months. According to another study, immune responses mostly formed after the second dose, and a 100% antibody seroconversion rate was observed following the third dose. In our study, the overall seroconversion rates to poliovirus types 1, 2, and 3 were 96% to 100% for the infants vaccinated with IPVAX™, Imovax Polio™, and Poliorix™. A randomized clinical trial in Puerto Rico showed 99–100% seroconversion rates among 230 infants who received IPVs (Sanofi-Pasteur, Lyon, France) at 2, 4, and 6 months of age [15]. Another Chilean study has shown that 98–100% recipients developed protective antibody titers to each poliovirus type after vaccination with IPVs at 2, 4, 6 months of age [16]. Similar results were achieved in Chinese infants receiving different schedules administered IPVs at 2, 3, and 4 months of age [17,18]. Primary three-dose vaccination with IPV-containing combined vaccine also demonstrated excellent immunogenicity and safety profiles in Taiwanese [19] and Guatemalan infants [20].

The strength of this study includes that we compared the immunogenicity and safety of eIPVs from different manufactures currently used in Korean infants. A potential limitation of this study was open-label design, which had no effect on immunogenicity assessment, but may have influenced safety profiles. Second, we did not assess the interchangeability of different eIPVs in the same participants. Third, fewer participants in IPVAX™ group have completed the study, and this might affect the power of the study. However, demographic characteristics, immunogenicity and safety profiles were similar in all three groups, and consistent with previous studies. Finally, in this study, we administered the single eIPV with DTaP vaccine concomitantly. However, in many countries, a combined vaccine including DTaP-IPV, DTaP-IPV/Hib, or DTaP-IPV/Hib/HepB was licensed and widely used. Such vaccines were approved based on the data that immunogenicity and safety of these combination vaccines were non-inferior compared to that of DTaP (Diphteria-Tetanus-acellular Pertussis), IPV or Hib (Haemophilus influenzae type b) vaccines administered separately. Therefore, it is unlikely that IPVs administered as a combined vaccine would generate different results from those in our study.

## 5. Conclusions

In conclusion, a three-dose primary vaccination with eIPVs from different manufacturers induced robust immune responses and had a clinically acceptable safety profile in healthy Korean infants. They are thought to be useful in preventing poliovirus infections if the primary vaccinations are completed according to the Korean NIP.

## Figures and Tables

**Table 1 vaccines-08-00200-t001:** Demographic characteristics.

Poliovirus Vaccine Type	IPVAX™ (N = 40)	Imovax Polio™ (N = 52)	Poliorix™ (N = 58)
Age at dose 1 (day)			
Mean ± sd (min, max)	63.80 ± 6.58 (44.00, 82.00)	64.65 ± 4.37 (59.00, 77.00)	61.21 ± 6.44 (46.00, 76.00)
Gender, *n*(%)			
Male	23 (57.50)	29 (55.77)	32 (55.17)
Female	17 (42.50)	23 (44.23)	26 (44.83)

N, number of participants; *n*(%), number(percentage) of participants in a given category; sd, standard deviation.

**Table 2 vaccines-08-00200-t002:** Seroconversion rates and neutralizing antibody titers after primary enhanced-potency inactivated poliovirus vaccine (eIPV) vaccination.

Dose Type	IPVAX™	Imovax Polio™	Poliorix™	*p*-Value
Type 1				
Seroconversion *n*/N (%)	40/40 (100.00)	51/52 (98.08)	57/58 (98.28)	0.6889
GMT (95% CI)				
Pre	6.50	5.73	4.96	
post	97.01	78.17	102.00	0.9591
	(75.55, 124.56)	(61.32, 99.64)	(74.82, 139.05)	
Type 2				
Seroconversion *n*/N (%)	40/40 (100.00)	50/52 (96.15)	58/58 (100.00)	0.1481
GMT (95% CI)				
Pre	5.37	7.89	5.66	
post	163.14	144.31	181.02	0.9747
	(131.42, 202.52)	(109.64, 189.96)	(141.02, 232.23)	
Type 3				
Seroconversion *n*/N (%)	40/40 (100.00)	50/52 (96.15)	58/58 (100.00)	0.1481
GMT (95% CI)				
Pre	5.10	5.89	5.39	
post	226.76	218.16	268.53	0.9763
	(180.27, 285.23)	(155.80, 305.48)	(207.59, 347.38)	

N, number of participants; *n*, number of participants in a given category; GMT, geometric mean titer; CI, confidence interval.

**Table 3 vaccines-08-00200-t003:** Incidence of adverse events after vaccination.

Heading	IPVAX™ (N = 40) *n* (%)	Imovax Polio™ (N = 52) *n* (%)	Poliorix™ (N = 58) *n* (%)	Total (N = 150) *n* (%)	*p*-Value
Dose #1					
Total AE	35 (87.50)	43 (82.69)	55 (94.83)	133 (88.67)	0.1293
Local AE	16 (40.00)	24 (46.15)	25 (43.10)	65 (43.33)	0.8392
Systemic AE	34 (85.00)	42 (80.77)	53 (91.38)	129 (86.00)	0.2713
Dose #2					
Total AE	34 (85.00)	43 (82.69)	55 (94.83)	132 (88.00)	0.1172
Local AE	18 (45.00)	17 (32.69)	18 (31.03)	53 (35.33)	0.3224
Systemic AE	31 (77.50)	40 (76.92)	55 (94.83)	126 (84.00)	0.0161
Dose #3					
Total AE	29 (72.50)	38 (73.08)	53 (91.38)	120 (80.00)	0.0217
Local AE	16 (40.00)	22 (42.31)	17 (29.31)	55 (36.67)	0.3238
Systemic AE	28 (70.00)	31 (59.62)	51 (87.93)	110 (73.33)	0.0031

AE, adverse event; N, number of participants; *n* (%), number (percentage) of participants in a given category; # means the standard primary series.

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
