# Peer review of "Comparison of the Immunogenicity and Safety of Three Enhanced Inactivated Poliovirus Vaccines from Different Manufacturers in Healthy Korean Infants: A Prospective Multicenter Study"

_vaccines, 2020, doi:10.3390/vaccines8020200_

Round 1

Reviewer 1 Report

In the manuscript entiteled: ”Immunogenicity and safety of enhanced inactivated Poliovirus vaccine in healthy Korean infants: A prospective multicenter study” by Kwak B.O et al. (manuscript vaccines-780709) the authors describe the results obtained from a clinical study on 150 children, where the immunogenicity and safety of three (five) inactivated polio vaccines are compared between May 2009 to April 2011. Participants were divided into three groups, with an initial 168 recruited participants where 46 obtained IPVAX, 60 participants received Imovax and 62 obtained PolioRix, all three enhanced Poliovirus vaccines administered through intramuscular or subcutaneous injections in a three-dose schedule (2, 4 and 6 months) and concomitantly given DTaP vaccine on the right side thigh. Blood was collected pre-vaccination and 4 to 6 weeks post final vaccination, thereafter AE and immunity as serological polio-neutralizing antibody titers were determined against the three poliovirus strains.

Comments and questions:

Q1. In Materials and Methods paragraph Statistical analyses, line 115 the authors state the following: “The required sample size was 53 in each group”.  Q1. In the IPVAX study group, only 40 individuals completed the study? The authors should comment this deviation from the initial plan and the reduced participant number and its impact on the statistical power of the study.

Q2. What was the reasons for the drop-out of participants?

Q3. In Table 1, the numbers of males and females are given, with a larger proportion of males in all study groups. The authors could comment on this selection, and also describe if there were any differences between side effects and in immunogenicity between genders in their material.

Q4. Table 1. In the IPVAX study group, only 40 individuals completed the study? Please comment why 40 individuals were accepted when 53 participants/group was aimed for in Materials and Methods section.

Q5. Table 2. The obtained Polio type 1 neutralization serotiter GMT was considerably lower than the neutralization GMTs against polio type 2 and polio type 3. Could the authors comment on why this difference is occur? (Will this mean that the neutralizing immunity will remain for a shorter time period against polio type 1 then the neutralizing titers against polio types 2 or 3 in the vaccine recipients?)

Author Response

We deeply appreciate reviewer 1 for critical reading of our manuscript and the comments.

Reviewer 2 Report

Immunogenicity and safety of enhanced inactivated poliovirus vaccine in healthy korean infants: a retrospective multicenter study 

Dear author and editor:

This study have confirmed the immunogenicity and safety of three poliovirus vaccines from different manufactureres in healthy Korean infants. the article is well written and could be published after a minor revision. 

i have some comments:

  • I think the authors could choose another title of the article and different from the title of the previous work (Kim JS et al, 2006). it is better to show a strong motive to study these three vaccines from different manufactureres. A previous study was developed to study the immunogenicity and safety of eIPV with some limitations, for this reason the aim of the study have to be clearly written. 
  • In the results section, the authors divided the results to (study population, immunogenicity and safety) which could be written in a better representative way for each point (e.g. AEs were similar between three groups).
  • In the discussion, i prefer to add more citations to compare the data with other international studies. 

Thank you very much, best regards 

Author Response

We deeply appreciate reviewer 2 for your kind suggestion and the comments on our manuscript.
